# Exploring the effect of lexical inferencing and dictionary consultation on undergraduate EFL students' vocabulary acquisition

Alaa Alahmadi[1], Anouschka Foltz[2]*

1 School of Languages, Literatures and Linguistics, Bangor University, Bangor, United Kingdom, 2 Institute of English Studies, University of Graz, Graz, Austria

* anouschka.foltz@uni-graz.at

## Abstract

This study compares how lexical inferencing and dictionary consultation affect L2 vocabulary acquisition. Sixty-one L1 Arabic undergraduates majoring in English language read target words in semi-authentic English reading materials and were either asked to guess their meaning or look it up in a dictionary. A pre- and delayed post-test measured participants' knowledge of target words and overall vocabulary size. The results show a significant and comparable learning effect for both vocabulary learning strategies (VLS), with a higher pre-test vocabulary size related to a larger learning effect for both VLS. In addition, the better participants were at guessing correctly, the better they learned words through inferencing. The results suggest that both VLS are equally effective for our learner group and that learners' overall vocabulary size influences the amount of learning that occurs when using these VLS.

## Introduction

Vocabulary acquisition is a fundamental aspect of foreign language learning that positively impacts the learning and communicative developments in that language [1] and that is vitally important for adequate and efficient communication [2]. Moreover, inadequate vocabulary knowledge can be a major obstacle in acquiring a target language [3]. Despite its value, vocabulary has not received the same attention that second language instructors and researchers have bestowed on other aspects of language, such as grammar and phonology. Lewis [4] criticized this neglect by stating that "language consists of grammatical lexis, not lexicalized grammar" (p. 95). Since then, research on vocabulary acquisition has increased [1,5–10] and instructors and course designers have increased their focus on how learners acquire and retain new words.

### Vocabulary learning strategies

A good number of foreign and second language learners adopt what is known as vocabulary learning strategies (VLS), either inside or outside of the classroom. Catalan [11] defined VLS

**Data Availability Statement:** The data and analysis scripts for RQ1 through RQ3 are available on the Open Science Framework at https://osf.io/zsvqk/.

**Funding:** The first author was funded by King Abdulaziz University, Kingdom of Saudi Arabia. The

authors also acknowledge the financial support by the University of Graz. The funders had no role in study design, data collection and analysis, decision to publish, or preparation of the manuscript.

**Competing interests:** The authors have declared that no competing interests exist.

as mechanisms that language learners typically apply to determine the meaning of unknown lexical items, retrieve them from long-term memory and apply them in verbal and written situations. The current study focuses on two particular VLS: guessing the meaning of words from context (lexical inferencing) and looking words up in a bilingual dictionary (dictionary consultation). We focus on these two strategies because they are among the most frequently used as reported by learners [9,12–15] and previous studies have yielded conflicting results in terms of the learning effect that both strategies yield [16–21].

Following Schmitt's [9] VLS taxonomy, both lexical inferencing and dictionary consultation are discovery strategies, which are employed to learn new words, rather than consolidation strategies, which are employed to remember initially learned words. Discovery strategies are further subdivided into determination strategies, which provide a set of limited choices to learn new words, and social strategies, which refer to the cooperation with others to learn new words. Both lexical inferencing and dictionary consultation constitute determination strategies. While learners use various VLS when reading a written text in the target language [22,23], some VLS seem to be more popular than others. As mentioned above, both dictionary consultation and lexical inferencing are rather popular [9,12–15], possibly because they are quite easily implemented.

## Processing depth

Many second language researchers assume that deeper processing and more elaboration of lexical information will increase retention compared to less deep processing and less elaboration [24]. This assumption is expressed in a number of psycholinguistic theories of vocabulary retention (e.g. [1,25–27]). Craik and Lockhart's [25] Level of Processing theory was one of the early theories on retention emphasising that learners could engage with novel materials either shallowly or in depth.

Hulstijn and Laufer's [26] more recent Involvement Load Hypothesis operationalises the concepts of processing depth and elaboration for L2 vocabulary learning. They introduce the notion of involvement load, made up of the components *need*, *search* and *evaluation*. Need is a motivational component that considers who has set the particular task of determining the meaning of a word. Need is moderate if an instructor sets this task for learners (e.g., by asking learners to look up a specific lexical item), and it is strong if learners set this task for themselves (e.g. deciding to look up a certain target word when reading a text). Search is a cognitive component that refers to whether the meanings of the new lexical items are given to learners (e.g. by providing them in the margins [28]) or whether they need to find them. Search is absent in the former case, and present in the latter. If search is present, it can be moderate if learners have to engage in receptive retrieval (e.g. looking for the meaning of a target word) or strong if learners have to engage in productive retrieval (e.g. determining the form of a target word) [26]. Finally, evaluation is a cognitive component that refers to whether learners have to compare or combine the new lexical items with other words. Evaluation is moderate when learners compare the meaning of the target lexical item with other words' meanings (e.g. comparing multiple meanings of the word *bank* against the provided context). Evaluation is strong when learners need to assess how the new word can combine with other words in a specific linguistic context (e.g. determining how the target word *intellectual* fits in with other words in an original sentence) [26]. The stronger these three components are, the greater is the involvement load and the better is the retention of the new lexical items [26]. To estimate involvement load for a task, scores are assigned to the strength of each component: 0 if a component is absent, 1 if it is moderate, and 2 if it is strong.

The Technique Feature Analysis is another more recent theoretical framework for lexical learning [27]. It involves five factors that affect the depth with which learners process new

lexical items that they encounter: *motivation*, *noticing*, *retrieval*, *generation* and *retention*. Each factor has associated questions about the particular task that learners are engaged in. Questions are posed such that a *yes* response captures a type of elaboration or processing that has been suggested to facilitate vocabulary learning (see [29]). For each question that can be answered with a *yes*, the task therefore receives 1 point for a possible total of 18 points. The complete set of questions from the Technique Feature Analysis is given in Table 1 in the following section.

## Lexical inferencing

Guessing or inferring meaning from context usually involves "direct mental operations" [30] (p. 117) to comprehend, categorize, store and memorize target words. Such lexical inferencing forms part of general reading skills [31] and is thought to enhance student autonomy in vocabulary acquisition [6,17]. It is generally assumed that a rather high amount of lexical text coverage is needed to successfully infer unknown lexical items from context [32]. Nation [33] suggests that 98% of text coverage is needed for adequate comprehension of a variety of written and spoken texts. This amount of coverage requires a vocabulary size in terms of breadth of between 8000 and 9000 word families for novels, newspaper articles and academic texts [31,33]. In contrast, simplified texts require knowledge of about 3000 word families for adequate comprehension [33].

Haastrup [34] suggests that inferencing includes "informed guesses as to the meaning of a word in the light of all available linguistic cues in combination with the learner's general knowledge of the world, her awareness of the co-text and her relevant linguistic knowledge" (p. 39). This involves three major sources of knowledge: Intralingual cues refer to knowledge from the target language; Interlingual cues refer to knowledge from the native language or any other foreign language; Contextual cues involve knowledge of the world and the linguistic

**Table 1. Technique Feature Analysis (adapted from [27], p. 7) with points assigned to the tasks of Lexical Inferencing (LI) and Dictionary Consultation (DC) as implemented in the current study during training.** For a detailed explanation of these questions and the concepts they refer to (see [29]).

| Factor | Questions | LI | DC |
|---|---|---|---|
| Motivation | Is there a clear vocabulary learning goal? | 1 | 1 |
| | Does the activity motivate learning? | 1 | 1 |
| | Do the learners select the words? | 0 | 0 |
| Noticing | Does the activity focus attention on the target words? | 1 | 1 |
| | Does the activity raise awareness of new vocabulary learning? | 1 | 1 |
| | Does the activity involve negotiation? | 1 | 1 |
| Retrieval | Does the activity involve retrieval of the word? | 1 | 1 |
| | Is it productive retrieval? | 0 | 0 |
| | Is it recall? | 1 | 0 |
| | Are there multiple retrievals of each word? | 0 | 0 |
| | Is there spacing between retrievals? | 1 | 1 |
| Generation | Does the activity involve generative use? | 1 | 1 |
| | Is it productive? | 0 | 0 |
| | Is there a marked change that involves the use of other words? | 0 | 0 |
| Retention | Does the activity ensure successful linking of form and meaning? | 0 | 1 |
| | Does the activity involve instantiation? | 1 | 1 |
| | Does the activity involve imaging? | 0 | 0 |
| | Does the activity avoid interference? | 1 | 1 |
| Total score out of a maximum of 18 | | 11 | 11 |

context. Inappropriate use of these cues might lead to incorrect guesses [35]. Some researchers suggest that the presence of clear textual cues besides the existing linguistic clues are vital components for correct lexical inferencing [36–38].

The literature has highlighted several factors which could influence the process of lexical inferencing. These factors fall into two main categories: contextual and learner-related aspects [39]. Contextual aspects include the importance of the unfamiliar word for text comprehension as a whole and the semantic richness of the context [40,41]. Learner-related factors comprise the level of attention that the learner dedicates to the provided text and the breadth and depth of his/her vocabulary knowledge [42,43]. The learner's level of engagement includes several factors, such as determining the meaning of surrounding vocabulary items, the grammatical structures, the structural character of surrounding sentences, the topic and the broader context of the target text [26].

In terms of processing depth and elaboration, the Involvement Load Hypothesis [26] assigns a high score of 4 out of a possible 6 to lexical inferencing as it is implemented in the current study, where participants were asked to read a text and guess the underlined words from context. Need is moderate and receives a score of 1 as learners were instructed to do the task, search is also moderate and receives a 1 as participants were engaged in receptive retrieval, and evaluation is high and receives a 2 as learners needed to establish how the word meaning fits into the particular linguistic context.

The Technique Feature Analysis [27] assigns lexical inferencing as implemented in current study a moderate to high score of 11 out of 18 (see Table 1; [44]). The task receives 2 points out of 3 for motivation as there is a clear vocabulary learning goal and the activity motivates learning. The task receives all 3 points for the noticing factor because the underlining focuses attention on the target words, the task directly involves new vocabulary learning, and inferencing involves negotiation of meaning (cf. [45]). Retrieval yields 3 points out of 5 since the target words need to be retrieved through recall and since there is spacing between the retrieval of successive target words. The task scores 1 out of 3 for generation as target words are encountered in a novel sentence. The task thus involves generative use. Finally, retention scores a 2 out of 4: The task involves instantiation as the written context can help with recall, and it mostly avoids interference as the task does not involve interference from semantic sets.

## Dictionary consultation

Many researchers (e.g. [46]) have noted the impact of dictionary consultation in general as a beneficial strategy for building learners' lexical knowledge. Dictionary consultation promotes reading comprehension and vocabulary acquisition (e.g. [47–49]). However, dictionary consultation might be an obstacle to the development of other VLS [50]. Specifically, learners relying mostly on basic and straightforward VLS strategies such as dictionary consultation may be reluctant to use more complicated strategies like lexical inferencing. Moreover, Nation [1] criticizes dictionary usage because it supports the notion that the first language (L1) has exact equivalences of the target language words.

There is no agreement on whether bilingual or monolingual dictionaries promote more successful vocabulary acquisition [51,52], with some studies highlighting more effective vocabulary acquisition through monolingual dictionary use [53], others through bilingual dictionary use [54], and yet others suggesting that both methods yield comparable outcomes [51]. Differences in learners' proficiency levels may have yielded these discrepant results. Specifically, bilingual dictionary usage is considered to be a beneficial strategy for less proficient learners as it encourages them to link the second language (L2) target word to their first language knowledge, allowing them to use the L1 as a reference to comprehend the L2 [55]. Furthermore,

Kroll and Curley [56] argue that new L2 vocabulary is effectively stored in the lexicon if it is linked to its L1 equivalent. In contrast, monolingual dictionaries may be more beneficial for more proficient learners [55].

Numerous researchers have discussed the depth of cognitive processing involved in dictionary consultation. For instance, Scholfield [57] argues that dictionary consultation involves exploring a target word's spelling, inflections and part of speech in addition to establishing the word's meaning, all of which contributes to vocabulary retention. In line with this, Liu et al. [58] suggest that dictionary consultation can involve deep cognitive processing (1) when considering unknown lexical features, (2) through repeated exposure to unfamiliar vocabularies, and (3) by detecting headwords and word syllables in dictionary content. In contrast, O'Malley and Chamot [59] argue that dictionary consultation involves a low degree of conceptual processing.

In terms of processing depth and elaboration, the Involvement Load Hypothesis [26] assigns a high score of 4 to dictionary consultation as implemented in the current study, where participants read a text and were asked to look up the meaning of underlined words in a bilingual dictionary. Specifically, need receives a 1 as learners were instructed to engage in dictionary consultation, search receives a 1 as learners engaged in receptive retrieval, and evaluation receives a 2 as learners needed to decide which of the suggested translations in the dictionary entry fits the sentence context.

The Feature Technique Analysis [27] assigns a moderate to high score of 11 out of 18 to dictionary consultation as implemented in the current study (see Table 1). Dictionary consultation scores identically to lexical inferencing, with two exceptions: In contrast to lexical inferencing, the type of word retrieval through dictionary consultation does not involve recall. While lexical inferencing involves recall, i.e. retrieving the meaning of target words from memory, dictionary consultation does not, but instead involves recognition of the correct translation among a number of choices. Furthermore, while lexical inferencing does not ensure that form and meaning are successfully linked because learners can guess incorrectly, dictionary consultation mostly allows for the successful linking of form and meaning.

## Vocabulary acquisition through lexical inferencing vs. dictionary consultation

The previous two sections have shown that the Involvement Load Hypothesis [26] and Feature Technique Analysis [27] assign the same number of points with a moderate to high score to lexical inferencing and dictionary consultation as they are implemented in the current study. These approaches thus predict that both strategies should yield a sizeable learning effect.

Numerous studies have explored how lexical inferencing and dictionary consultation affect vocabulary acquisition. Shangarfam et al. [17] examined the impact of bilingual dictionary consultation and guessing from context on learners' ability to select the appropriate vocabulary item in a fill-in the blank exercise. Participants were explicitly taught inferencing procedures and finding a word's meaning in a bilingual dictionary. The pre- and post-tests were multiple choice fill-in-the-blank exercises, where participants had to decide which of four words correctly fills the blank. The results indicated that the context-guessing group significantly outperformed the dictionary look-up group in terms of selecting the correct vocabulary item in the post-test fill-in-the-blank task. This suggests that practicing inferencing strategies increases performance at tasks that require guessing meaning from context.

In contrast, other studies found better vocabulary acquisition through dictionary consultation (mostly using bilingual dictionaries) than through guessing (e.g. [16,19,60,61]). Participants in Zou [19] read a text, either translated ten underlined words using a monolingual

dictionary or inferred the meaning of the underlined words, and answered comprehension questions about the text. Participants were tested on the meanings of the underlined words in both immediate and delayed post-tests. Results showed better vocabulary acquisition for the dictionary consultation strategy compared to inferencing.

Still other studies found that both strategies similarly assisted learners' vocabulary acquisition (e.g. [18,62]). Participants in Mondria [63] either looked up the translation of target words in a provided word list or guessed the meaning of target words from context and then verified their guess through the provided words list (to prevent learning incorrect meanings for target words). Participants in both groups had similar vocabulary retention levels, but the guessing-plus-verifying method was significantly more time consuming than the look-up strategy.

## How lexical inferencing and dictionary consultation relate to vocabulary size

Several studies have found relationships between the frequency of using lexical inferencing and dictionary consultation strategies and learners' vocabulary size (e.g. [39,64]). For example, Alahmadi and colleagues [65] found a significant positive relationship between learners' vocabulary size and how frequently learners reported guessing the meaning of words from context. In the same vein, Gu and Johnson [64] found that contextual inferencing and dictionary use positively correlated with participants' vocabulary knowledge. While it is likely that increased dictionary use leads to a larger vocabulary size rather than the other way around, the same is not necessarily the case for lexical inferencing. Specifically, it is possible that successfully guessing word meaning from context increases learners' vocabulary size, but it is also conceivable that learners with larger vocabularies use this strategy successfully more often than learners with smaller vocabularies. Specifically, learners with larger vocabulary sizes have greater lexical coverage and may thus be able to engage in lexical inferencing more successfully [33].

## The current study

This study examines whether lexical inferencing or dictionary consultation supports initial learning and retention of English vocabulary among male and female Saudi senior undergraduate English-major students. We exposed students to target words in semi-authentic reading materials and measured their knowledge of these target words and their overall vocabulary knowledge in terms of breadth prior to and following exposure. During the exposure phase, students were asked to guess the meaning of some of the target words and to look up the meaning for others in a dictionary. The pre- and delayed post-tests allowed us to calculate a learning effect in both the guessing and dictionary conditions. We explored whether these two vocabulary learning strategies impact target word acquisition. The study contributes to the currently rather heterogeneous picture of how lexical inferencing and dictionary consultation contribute to vocabulary acquisition. Unlike many previous studies, we used a within-participant design and a comparatively authentic learning situation, thus emphasizing ecological validity [66]. Furthermore, our design allowed us to measure the amount of successful lexical inferencing. This study intends to answer the following research questions (RQs):

1. Do learners show a larger learning effect for words that were trained in the two training sessions than for words that were not trained?

2. Do learners show a larger learning effect for words that they guessed in the two training sessions than for words that they looked up in a dictionary?

3. Does learners' vocabulary size, previous knowledge of the trained words, success in guessing, success in correctly looking up words and/or success in correctly answering comprehension questions about the texts influence how large their learning effect is for (a) words that they guessed and (b) looked up in the two training sessions?

## Methodology

### Participants

Sixty-one Saudi senior undergraduate English major students (47 [77%] males; 14 [23%] females) from three different Saudi Universities participated. Participants' ages ranged from 20 to 28 years (M = 22.75; *SD* = 1.63). All participants had Arabic as their first language. They started learning English in grade 4 of primary school and had received approximately 1600 hours of EFL instruction between their public school and university education [67]. Two additional participants were excluded because they did not guess or did not look up any target words during the training sessions (see below).

### Materials and procedure

The study was approved by the ethics committee of the College of Arts, Humanities and Business at Bangor University (approval number LX-1610). The study involved four sessions, a pre-test, two training sessions and a delayed post-test (see Fig 1). During the pre-test, participants completed (a) an English language self-assessment questionnaire gauging their use and knowledge of English, (b) Al-Masrai and Milton's [68] vocabulary size test XK_Lex, which provides an estimate of learners' breadth of lexical knowledge out of the most common 10,000 words in English, and (c) a translation task gauging participants' knowledge of 48 words that were relevant for the current study. During the two training sessions, participants read four texts (two per test session) that included a vocabulary task and comprehension questions. During the delayed post-test, participants completed (a) a vocabulary learning strategy (VLS) questionnaire, (b) the XK_Lex test again, and (c) the translation task again.

**Pre-test.** *English-language self-assessment questionnaire.* An English-language self-assessment questionnaire, given in Arabic, assessed how participants rated their English knowledge in the four language skills (listening, speaking, reading and writing). The questionnaire recorded demographic information (age, native language). Learners also rated their English generally as near native (4), fluent (3), advanced (2), intermediate (1) or beginner (0) and gauged how frequently they used English outside the classroom using the Likert scale: always (4), frequently (3), sometimes (2), rarely (1) or never (0). Participants then gauged how frequently various statements about reading, writing, listening, and speaking applied to them, using the same Likert scale.

*Word translation task.* A word translation task before and after training assessed participants' pre-test and delayed post-test knowledge of the words trained during the training sessions. The translation task consisted of 24 target words, which were trained during the main

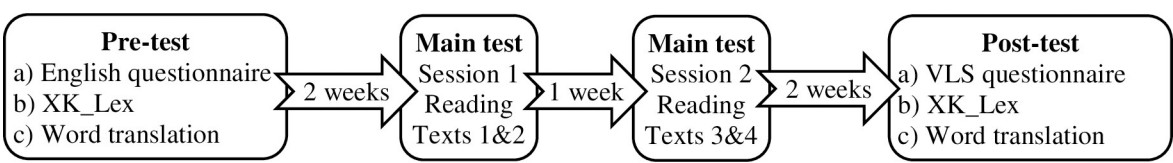

**Fig 1. The overall study design.**

training sessions, and 24 control words, which were not trained during the main training sessions. None of the target or control words were cognates with or loanwords from Arabic (see S1 Appendix for a list of target and control words).

Target and control words were matched for frequency, difficulty, word length, derivational complexity and part of speech. Target and control words had mean frequencies of 1764 words (SD = 2685) and 1745 words (SD = 1898), respectively, in the BYU British National Corpus (BYU-BNC; [69]), which did not differ statistically significantly (generalized linear model with family = "quasipoisson" for overdispersed count data: $\beta$ = -0.01, SE = 0.38, z = -0.03, p = 0.98).

A norming study with 16 senior undergraduate English major Saudi students (11 females, 5 males; mean age = 22.38, SD = 3.03; self-assessed proficiency level of 2.43, SD = 0.72) assessed the target and control words' difficulty levels. Students provided Arabic translations for the target and control words without consulting dictionaries or any other aids, and we counted how many points participants scored in the translation task. Scoring followed Wesche and Paribakht's [38] system, with participants receiving 1 point for presenting a semantically and syntactically suitable translation, 0.5 points for a partial success, such as providing an incomplete meaning or a semantically but not syntactically applicable response, and 0 points for an incorrect translation. This means that the maximum possible score corresponded to the number of words scored, and scores therefore approximated the number of words known. Participants correctly translated an average of 2.40 (SD = 1.86) and 2.13 (SD = 2.06) of the 24 target and 24 control words, respectively, which did not differ statistically significantly (generalized linear model with family = "poisson" for count data: $\beta$ = 0.12, SE = 0.19, z = 0.62, p = 0.53).

Letter counts revealed that target and control words had mean word lengths of 8.96 letters (SD = 1.90) and 7.86 letters (SD = 3.05), respectively, which did not differ statistically significantly (generalized linear model with family = "poisson" for count data: $\beta$ = 0.13, SE = 0.10, z = 1.29, p = 0.20). Target and control words had mean derivational complexities (as number of derivational affixes) of 0.58 (SD = 0.65) and 0.42 (SD = 0.50), respectively, which did not differ statistically significantly (generalized linear model with family = "poisson" for count data: $\beta$ = 0.34, SE = 0.41, z = 0.81, p = 0.42). Finally, there was no significant difference in how the parts of speech noun, verb, adjective and adverb were distributed across the target and control word lists (Pearson's Chi-squared: $\chi^2$ = 3.82, df = 3, p = 0.28).

Overall, this suggests that target and control words are quite well matched for frequency, difficulty, word length, derivational complexity and part of speech. It needs to be noted though that the absence of statistically significant differences in the above tests does not provide evidence for equivalence [70]. Instead, matching merely ensures that target and control words are relatively comparable across a range of factors. Importantly for the current study, participants in the norming study correctly translated only a minority of target and control words, suggesting that their knowledge of these words is not already at ceiling.

*The XK_Lex vocabulary size test.* The XK_Lex test [68] assessed participants' vocabulary knowledge in terms of breadth out of the most familiar 10,000 words in English. The test estimates the number of lemmas, i.e. the number of headwords and some inflected and reduced forms [1], that learners know. For instance, the lemma for the word *perform* includes *performs*, *performed* and *performing*, but not *performer*. Participants selected the words they knew from a total of 120 words spread across ten columns with twelve words per column. Each column contained ten real words and two pseudo-words (to minimise the influence of guessing on learners' responses) for a total of 100 real words and 20 pseudo words. Participants' vocabulary size (as number of lemmas) was calculated using the following formula.

$$\text{vocab size} = \sum \text{selected real words} * 100 \ - \sum \text{selected pseudo words} * 500 \quad (1)$$

We selected the XK_Lex test for numerous reasons: It is considered to be a reliable and valid test to measure participants' breadth of lexical knowledge [68] and has been used in various previous studies (e.g. [71–73]). It is quick and easy to administer [74], taking only about 5 to 10 minutes. It is therefore less time-consuming than, for example, the Vocabulary Levels Test (VLT; [75], revised by [76]) or the Vocabulary Size Test (VST; [77]), which take 30 to 45 minutes to administer [78,79]. Since we were testing university students, we expected some students to have vocabulary sizes above 5,000, but below 10,000. We therefore chose a test that covers all frequency bands from 1,000 to 10,000 [68] as opposed to, for example, the VLT [76], which skips bands 6,000 through 9,000.

We also chose the XK_Lex because it is less likely to overestimate vocabulary knowledge than some other tests: First, its unit of word count is the lemma, as opposed to the word family used in many other vocabulary knowledge tests, including the VLT [76] and VST [77]. Lemmas are less likely to overestimate vocabulary knowledge because learners can typically easily derive inflected forms from the headword, but may not know all members of a word family [80], which tests based on the word family assume. Furthermore, the multiple choice format of the VLT [76] and VST [77] may overestimate learners' vocabulary size [81]. In contrast, the XK_Lex [68] includes pseudo-words to control for the potential amount of guessing.

That said, all vocabulary size tests have their limitations. Most importantly, the XK_Lex involves only self-report of word knowledge, not an actual demonstration of word knowledge. This means that learners' personal characteristics may affect the results of the XK_Lex substantially more than tests that require demonstration of knowledge. We will therefore address how personality traits may have influenced the results from the XK_Lex test in the discussion section.

*Pre-test Procedures*. Participants were tested in their classrooms at three Saudi universities. They gave informed consent immediately before the pre-test. As part of this, they could consent to grant the researcher access to their academic Grade Point Average (GPA), which was used as a general estimate of their English language academic performance [82,83], reflecting long-term monitoring of students' language level in a degree program with more than 95% of academic classes delivered in English.

After giving consent, students completed the English-language self-assessment questionnaire, the word translation task, and the XK_Lex vocabulary size test. Participants' were then grouped into a low and high English proficiency level. Specifically, each participant whose scores for the GPA, translation task and vocabulary size test were below the median scores for two or more of these measures was considered to have low English proficiency and vice versa.

**Training.** *Counterbalancing*. Participants were distributed across two groups (A and B). Assignment to groups was not entirely random. Instead, participants with low English proficiency were randomly distributed across these two groups; separately, participants with high English proficiency were randomly distributed across both groups. This was done to ensure a spread of abilities across both groups, with half the participants in each group having low vs. high English proficiency, respectively. We refrained from an entirely random assignment of participants to minimize proficiency differences across the two groups. Specifically, we wanted to avoid a situation where the majority of participants in one group have high English proficiency, whereas the majority of participants in the other group have low English proficiency. Text order and task (lexical inferencing or dictionary consultation) during the two training sessions were also counterbalanced across the two groups, as illustrated in Table 2. Notice that this counterbalancing ensured that both groups engaged in the guessing tasks and in the dictionary tasks at the same time, e.g. while one group engaged in guessing with Text 1, the other group also engaged in guessing, but with Text 2. This allowed us to better check that participants were actually following instructions. Specifically, during guessing tasks, the experimenters ensured that participants in both groups were not using a dictionary. During dictionary

Table 2. Overview of the study's counterbalancing.

| Groups | A | B |
| --- | --- | --- |
| Session 1 | Task 1: Text 1 Guessing | Task 1: Text 2 Guessing |
| | Task 2: Text 2 Dictionary | Task 2: Text 1 Dictionary |
| Session 2 | Task 1: Text 3 Dictionary | Task 1: Text 4 Dictionary |
| | Task 2: Text 4 Guessing | Task 2: Text 3 Guessing |

tasks, the experimenters observed that learners were frequently using a bilingual dictionary of their choice, mostly through apps on their phones. While we cannot entirely rule out that some participants may have guessed some words during a dictionary task, the frequent dictionary use suggests that, if this happened, it should have been a rather rare occurrence.

*Reading Texts*. The four texts used for training were adapted from two primary English textbooks [84,85]. All four texts had a moderate level of difficulty, especially regarding vocabulary, which we considered to be well suited for undergraduate students majoring in English. Individual vocabulary items that were deemed too specialized (for example, medical terminology) or culturally inappropriate were replaced with more appropriate words. All texts had similar lengths, which was achieved by shortening longer texts. The vocabulary profiler software Lextutor [86] determined that knowledge of 3,000 word families were needed for 95% text coverage and of 4,000 word families for 98% text coverage for the four texts. In addition, only two words were beyond the 5,000 word family level.

Six words in each text, corresponding to the target words in the pre- and delayed post-test word translation tasks, were underlined (4 texts x 6 words = 24 target words). Students were asked to provide the Arabic meaning for each underlined word. For two texts, they translated all the underlined words that they knew, and looked up the remaining words in a dictionary. For the remaining two texts, participants translated all the underlined words that they knew, and guessed the meaning of the remaining words from context. In both cases, participants engaged in translation from the L2 to the L1, which is generally considered to be easier for learners than translation from the L1 to the L2 [87]. Two different response columns were provided, and participants were asked to write their translation in the first column if they already knew the word and in the second column if they had looked up or guessed the word. Each text was followed by two multiple-choice text-comprehension questions. There was no time restriction to complete the task, but most participants completed it in about 20 minutes. Participants completed the two training sessions two weeks and three weeks, respectively, after the pre-test.

*Scoring*. The first author rated participants' translations of underlined words in line with Wesche and Paribakht's [38] scoring system. As in the pilot study, participants received 1 point for presenting a semantically and syntactically suitable translation, 0.5 points for a partial success, and 0 points for an incorrect translation. A second Arabic-English bilingual coder rated translations from a random subset of 20 participants. Inter-coder agreement was extremely high with 98.75% agreement (Cohen's Kappa $\kappa = 0.987$; $p < 0.001$).

**Delayed post-test.** The delayed post-test occurred two weeks after the second main training session. Participants completed the word translation task, the XK_Lex vocabulary size test, and a vocabulary learning strategy questionnaire. Results from the latter two are beyond the scope of this paper, but we mention these instruments here for reasons of transparency.

## Results

The data and analysis scripts for RQ1 through RQ3 are available on the Open Science Framework at https://osf.io/zsvqk/.

## Participants' English language profile

Participants rated their English-language proficiency as, on average, 2.32 (*SD* = 0.71), a value between *advanced* (2) and *fluent* (3). Participants' average vocabulary size during the pre-test of 3331 words (*SD* = 1318), however, is substantially lower than what has been suggested for high text coverage for authentic spoken and written materials [31,33]. It is therefore possible that participants overestimated their proficiency, and that they might only be at the intermediate level. Participants rated their English use outside of the classroom with a mean value of 1.97 (*SD* = 0.98), roughly corresponding to *sometimes* (2). Table 3 presents participants' mean scores from the statements that self-assessed their English reading, writing, listening and speaking. Most of the rating values for the four language skills are between *sometimes* (2) and *frequently* (3).

## Target word learning (RQ1)

We first report participants' knowledge of the target and control words before and after the training. As expected, before training participants knew only a minority of the 24 target (M = 4.61, *SD* = 4.40) and 24 control (M = 3.75, *SD* = 3.39) words. This number of known words was larger than expected based on the norming study, where participants scored, on average, below 3 for both the target and control words. Compared to the norming study, there is thus a somewhat larger than expected proportion of target and control words that cannot be learned because they are already known. Nevertheless, the vast majority of words are unknown, avoiding a ceiling effect and leaving room for additional learning during training. In line with the number of target words known before training, participants also reported knowing only a minority of target words during training (M = 6.56, *SD* = 4.85), with slightly lower numbers when adjusted for accuracy, i.e. for words that participants reported as knowing and that were also scored as correct (M = 5.11, *SD* = 4.58). After training, participants had substantially increased their knowledge of the 24 target words (M = 9.97, *SD* = 6.05), but numerically less so for the 24 control words (M = 5.43, *SD* = 3.71).

We then assessed whether the training sessions yielded a learning effect at all by analysing whether participants showed more learning for target words compared to control words. Fig 2 shows the mean vocabulary learning effect, i.e. the average increase in correctly translated words from pre-test to delayed post-test, for target and control words. This was calculated by subtracting the score for correctly translated words in the pre-test from that in the delayed

**Table 3. Mean scores of English language self-assessment questionnaire.**

| Self-Assessment statements | M *(SD)* |
|---|---|
| I recognise the main ideas when reading texts in my course textbooks | 2.76 (*1.00*) |
| I can locate the information that I need in a general text in a quick and easy manner | 2.65 (*0.88*) |
| I can comfortably read complex lengthy texts, stories and articles | 1.84 (*1.19*) |
| I can freely write my opinion on a variety of topics | 2.29 (*0.94*) |
| I can take notes during lectures | 2.49 (*1.03*) |
| I can build up my arguments in a logical way within an essay | 1.62 (*1.16*) |
| I can understand informal conversations on common topics | 2.76 (*0.93*) |
| I can easily follow lectures and presentations when they are conveyed clearly | 2.90 (*0.82*) |
| I can understand the news on the radio or TV | 2.35 (*0.99*) |
| I can express myself confidently within informal life situations | 2.68 (*0.86*) |
| I can present an academic topic in front of my class | 2.03 (*1.24*) |
| I can participate in an academic argument during lectures | 2.11 (*1.06*) |

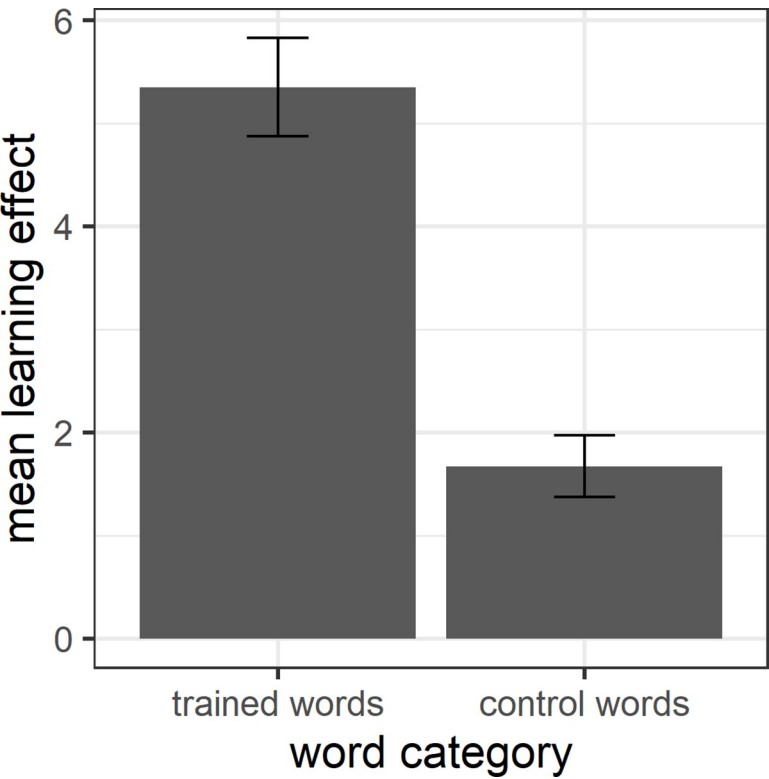

**Fig 2. Mean learning effect for trained and control words.**

post-test. Importantly, our calculation of the learning effect takes into account the two groups' pre-knowledge of the target words, i.e. their knowledge before training, and therefore rules out that the learning effect is merely due to the two groups differing in their knowledge of the target words before the training sessions. A Welch two sample t-test showed that the mean learning effect is significantly higher for trained words than control words ($t = 6.54$, $df = 100.45$, $p < 0.001$). Cohen's $d = 1.18$ (confidence interval: 0.80–1.57) shows a large effect size.

We then conducted two additional analyses focusing on the target words only. First, a pairwise comparison analysis gauged how the number of target words self-reported as known (i.e. words that participants translated in the response column for known words during training) and actually known (i.e. words that participants *correctly* translated in the response column for known words) during the training session compared to the number of target words known in the pre- and delayed post-tests. Pairwise comparisons were calculated from a mixed effects model using a Poisson distribution for count data with number of known words as the dependent variable and context (self-reported-as-known during training, actually known during training, pre-test, and delayed post-test), i.e. the context in which word knowledge was measured, as the independent variable. Random intercepts for participants were also included in the model. The results are shown in Table 4, and confirm that participants' knowledge of target words was significantly higher in the delayed post-test not only compared to the pre-test, but also compared to words reported as known and words actually known during training. Additionally, target word knowledge for the pre-test and actual knowledge during training did not differ significantly, suggesting that while learning did occur from training to delayed post-test, we find no evidence for learning from pre-test to training. Finally, participants' self-reported knowledge of target words during training was significantly higher than their actual

**Table 4. Post-hoc tests for knowledge of target words across different contexts.**

| contexts | β | SE | t | p |
|---|---|---|---|---|
| pre-test vs. self-reported as known | -0.35 | 0.05 | -6.41 | < 0.001 |
| pre-test vs. actually known | -0.10 | 0.06 | -1.78 | = 0.28 |
| pre-test vs. delayed post-test | -0.77 | 0.05 | -15.17 | < 0.001 |
| self-reported-as-known vs. actually known | 0.25 | 0.05 | 4.67 | < 0.001 |
| self-reported-as-known vs. delayed post-test | -0.42 | 0.05 | -9.24 | < 0.001 |
| actually known vs. delayed post-test | -0.67 | 0.05 | -13.61 | < 0.001 |

knowledge of target words for both the pre-test and during training, indicating that they significantly over-reported their word knowledge during training.

Second, a correlation analysis shows that the number of target words self-reported as known and actually known during training is highly correlated (t = 22.54, df = 59, p < 0.001, $R^2$ = 0.9), with 90% of the variability in self-reported knowledge explained by the variability in actual knowledge.

## Word learning through lexical inferencing vs. dictionary consultation (RQ2)

We first report participants' knowledge of the target words learned through inferencing vs. dictionary consultation before and after the training. Before training, participants received similar scores for the 12 target words that they would be asked to guess (M = 2.24, *SD* = 2.41) and for the 12 target words that they would be asked to look up (M = 2.38, *SD* = 2.36) during training. After training, participants had substantially increased their knowledge of both the 12 target words that they were asked to guess (M = 5.04, *SD* = 3.04) and the 12 target words that they were asked to look up (M = 4.93, *SD* = 3.52).

We then tested whether participants showed a larger learning effect for words that they guessed than for words that they looked up in a dictionary. Fig 3 shows the mean vocabulary learning effect for the two learning conditions, calculated as above to control for participants' pre-knowledge of target words. A paired t-test found no evidence that guessing words from context yields a significantly larger learning effect than looking words up in a dictionary (*t* = 0.59, *df* = 119.85, *p* = 0.55). In line with this, Cohen's *d* = 0.11 (confidence interval: -0.25– 0.47) is negligible.

## Factors influencing word learning (RQ3)

Next, two separate generalized linear models explored which factors influenced the mean vocabulary learning effect (a) for words that participants guessed and (b) for words they looked up in the dictionary. The independent variables were participants' vocabulary size (pre-test score from the XK_Lex; numeric from 0 to 10000), previous knowledge of the trained words (pre-test score for target words from the translation task, numeric from 0 to 24), success in guessing during training (score for correctly guessed words divided by the total number of words guessed; numeric from 0 to 1), success in correctly looking up words during training (score for correctly looked-up words divided by the total number of looked-up words; numeric from 0 to 1), and success in correctly answering the comprehension questions during training (number of correctly answered comprehension questions; numeric from 0 to 8). The dependent variable for the first model was the mean vocabulary learning effect for words that participants encountered in the guessing condition during training (delayed-post-test minus pre-test score for guessed words from the translation task; numeric from -12 to 12); The dependent

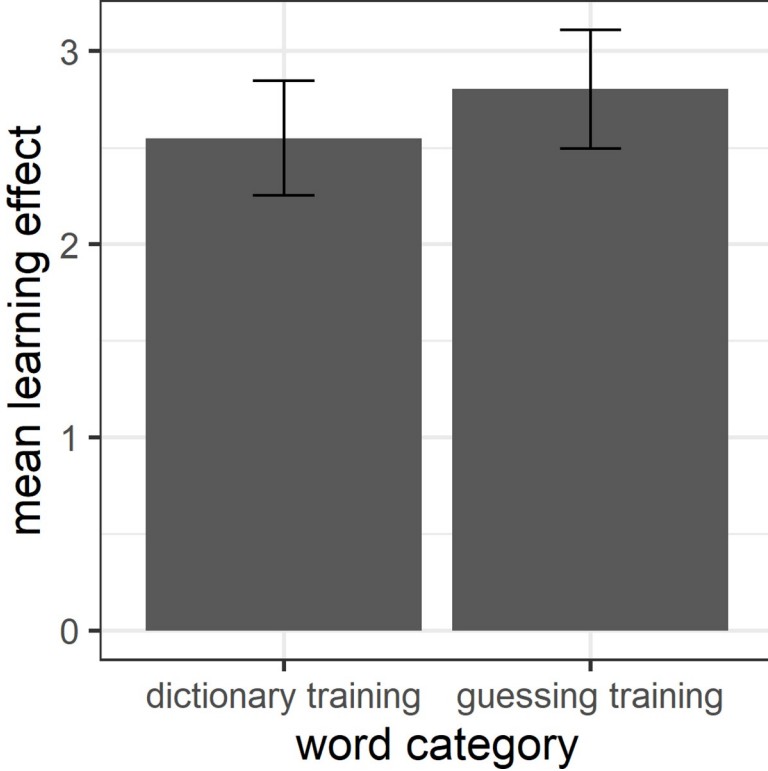

**Fig 3. Mean learning effect for dictionary training and guessing training.**

variable for the second model was the mean vocabulary learning effect for words that partici-
pants looked up during training (delayed-post-test minus pre-test score for looked-up words
from the translation task; numeric from -12 to 12).

All independent variables were centred prior to analysis to minimize collinearity. Indepen-
dent variables that did not significantly contribute to model fit were removed from the models
in a step-wise procedure to yield the final statistical models. The model for the guessing condi-
tion revealed three significant main effects, showing that the higher participants' pre-test
vocabulary size ($\beta = 0.001$; $SE = 0.0006$; $t = 2.03$; $p < 0.05$) and the better participants were at
guessing correctly ($\beta = 2.39$; $SE = 1.03$; $t = 2.33$; $p < 0.05$), the better they learned the words in
the guessing condition. Furthermore, the more words participants already knew before train-
ing, the lower their learning effect in the guessing condition ($\beta = -0.22$; $SE = 0.09$; $t = -2.46$;
$p < 0.05$). The marginal R-squared [88] for the model is $R^2_m = 0.17$, suggesting that 17% of the
variance in the data can be explained by the fixed factors.

Various authors have emphasized the need for large lexical coverage, which requires a suffi-
ciently high vocabulary size to be able to understand a text and infer unknown words success-
fully [33]. We therefore conducted an additional analysis to test for a possible relationship
between vocabulary size and guessing. This additional generalized linear model had vocabu-
lary size as independent variable and success in guessing as dependent variable. The results
showed that participants with higher vocabulary knowledge were significantly more successful
at guessing from context than participants with lower vocabulary knowledge ($\beta = 0.0001$;
$SE = 0.00006$; $t = 2.89$; $p < 0.01$).

The final statistical model for the look-up condition revealed only one main effect showing
that the higher participants' vocabulary size, the better they learned the words in the look-up

condition ($\beta = 0.001$; $SE = 0.0004$; $t = 3.00$; $p < 0.01$). The marginal R-squared for the model is $R^2_m = 0.13$, suggesting that 13% of the variance in the data can be explained by participants' vocabulary size.

## Discussion

The current study found a clear learning effect for the words trained in the study, with a statistically significantly higher learning effect for target words compared to control words. Words trained through lexical inferencing and dictionary consultation produced similar learning effects, suggesting that both learning methods are equally effective for the current learner group. Learners' vocabulary size predicted the size of their learning effect in the guessing and look-up conditions. Specifically, the larger participants' vocabulary size, the larger their learning effect. The learning effect in the guessing condition was additionally influenced by the number of words that participants already knew during training, with higher previous knowledge related to lower learning effects, and by participants' success in guessing correctly, with participants who were better at guessing correctly showing a larger learning effect.

### Target word learning

The first research question (RQ1) investigated whether learners' retention in the delayed post-test for words that were trained in the two training sessions was better than their retention for words that were not trained. Our results indicate that this was indeed the case. This suggests that encountering a word while reading for comprehension and engaging with the word by either guessing its meaning from context or looking it up in a dictionary yields a larger learning effect than vocabulary learning that would have occurred anyway during the duration of the study. The trajectory of actual and self-reported target word knowledge throughout the study confirmed the significant learning effect: Participants knew significantly more target words in the delayed post-test than in the pre-test and during training. Even though participants' self-reported knowledge of target words during training overestimated their actual knowledge of target words both during training and in the pre-test, their actual target word knowledge in the delayed post-test still significantly exceeded their self-reported knowledge during training. It is thus not merely the case that learners retained the same amount of knowledge that they reported during training. Instead delayed post-test knowledge of target words significantly exceeds all three prior measures of target word knowledge. These findings emphasize the substantial role of the examined VLS (lexical inferencing and dictionary consultation) on vocabulary learning.

The current results suggest that the trained words were processed in a sufficiently deep and meaningful manner within the context in which they appeared, resulting in enhanced retention. In contrast, most control words were only encountered in isolation and without any contextual cues for learning [89] during the pre- and delayed post-tests, resulting in significantly less retention. Generally, rich and deep semantic processing is believed to facilitate the learning process [90]. The current results are also in line with prior-mentioned studies in which lexical inferencing and dictionary consultation are considered as strong predictors of vocabulary retention [18,63].

### Word learning through lexical inferencing vs. dictionary consultation

Our second research question (RQ2) explored whether a larger learning effect would be observed for words that participants guessed in the two training sessions compared to words that they looked up in a dictionary. Results revealed a similar learning effect across the two learning situations. These findings are consistent with Çiftçi and Üster's [62], Mondria's [63]

and Zaid's [18] outcomes, in which lexical inferencing produced similar levels of retention as dictionary consultation. It seems that both VLS play similar roles in building learners' vocabulary knowledge.

This result is also consistent with the predictions of the Involvement Load Hypothesis [26] and Technique Feature Analysis [27]. Both approaches suggest similar processing depth and elaboration for lexical inferencing and dictionary consultation, as implemented in the current study. In line with this, both VLS showed a comparable learning effect. In fact, the similar levels of processing depth of both VLS according to the Involvement Load Hypothesis and Technique Feature Analysis might be a possible explanation for the finding that both strategies led to similar levels of word learning and retention. Furthermore, both approaches suggest that lexical inferencing and dictionary consultation, again as implemented in the current study, involve relatively deep processing and elaboration. In line with this, participants showed a significant learning effect for target words compared to control words even though they encountered each target word only once during training. Engagement with each target word during this single encounter sufficed to yield a measurable learning effect.

However, our results are not consistent with some studies that have highlighted the superiority of one strategy over the other in terms of impact on learners' vocabulary retention level. For instance, Akpınar et al. [20], Shokouhi and Askari [91] and Shangarfam et al. [17] found that inferencing is more efficient than dictionary use. Most studies linked such findings to the deep mental processes involved in inferencing, as a learner will need a combination of cognitive techniques, linguistic clues and world knowledge to determine the meaning of a word. According to Van Parreren [92], inferencing engages many mental processes, such as determining the word form and linking current context with one's background knowledge. Such cognitively meaningfully processed materials enrich retention [93].

Other studies highlighted the role of dictionary consultation as a strong predictor of learners' vocabulary retention (e.g. [16,19,61]). Dictionary use often provides different aspects of word knowledge besides the meaning, for instance, the word's pronunciation, synonyms, derivatives and example sentences or phrases [19]. All this information can establish a cognitive network or foothold for the target word in the learner's mind.

While both lexical inferencing and dictionary consultation involve a fair amount of cognitive engagement, the kind of engagement seems to differ across the two VLS. For example, Amirian and Momeni [16] found that learners who applied guessing techniques paid more attention to word roots and meaning than morphological and phonological features. Moreover, successful lexical inferencing usually requires a high L2 proficiency level and an adequate vocabulary size [34,42,43], whereas even beginning learners can successfully use a bilingual dictionary. Interestingly, participants in the current study benefited equally from dictionary consultation and lexical inferencing, even though their average vocabulary size of 3331 was somewhat below the suggested 98% text coverage needed for adequate comprehension and successful lexical inferencing. Specifically, knowledge of about 4000 word families is needed for 98% text coverage for the four texts used in the current study, with 3000 word families yielding only 95% text coverage, which is likely to lead to adequate text comprehension only in a small minority of learners [94]. This suggests that learners whose vocabulary sizes allow them less than 98% of text coverage can successfully engage in lexical inferencing.

Our findings indicate that both VLS methods lead to higher learning than the baseline, but we found no evidence that inferencing lead to higher learning than dictionary use or vice versa. Discrepancies of our results with previous findings could be linked to many factors, such as the applied methodological approach, learners' vocabulary size, learning styles, proficiency levels or learners' motivation to participate in the study. For example, Shangarfam et al. [17] differs from the current study in relevant ways. While VLS was a within-participant factor

in the current study, Shangarfam et al. [17] used a between-participant design, where each participant engaged in either lexical inferencing or dictionary consultation (see e.g. [19,93], for other studies where VLS was a between-participant factor). Thus, Shangarfam et al. [17] may have found higher learning from inferencing than dictionary use due to individual differences in participants' retention abilities or engagement with the task. Furthermore, Shangarfam et al. [17] explicitly taught participants how to apply these two strategies. One possibility is therefore that inferencing leads to higher learning than dictionary use, but only if students are aware of or have been trained in proper inferencing strategies. There is, however, evidence that not all inferencing strategies need to be explicitly taught. For example, using a think-aloud protocol, Yayli [95] found that learners naturally used some guessing strategies, for example cohesive ties, such as anaphora, conjunctions, causal cohesion etc., as clues to guess word meaning. In addition, not all studies have found that inferencing strategies lead to higher retention than dictionary consultation even when participants are taught inferencing strategies. For example, Amirian and Momeni [16] found that high school students who were explicitly told the target words' meanings and experienced the words in suitable contexts outperformed students who were taught how to infer target words' meanings from context. One possible explanation for this finding is that participants, particularly at lower proficiency levels, may not successfully use inferencing strategies even if explicitly taught. Alternatively, experiencing target words in meaningful sentences after being provided the correct translation might have impacted the words' recall.

Azin et al. [93] also found higher learning for inferencing over dictionary consultation. Their control group was explicitly taught the target words' meanings, whereas their experimental group was asked to infer the target words from provided context. The experimental group additionally had the opportunity to verify their guess by using a dictionary. The differences in results between Azin et al. [93] and the current study may be related to this additional verification method. For example, it is possible that guessing followed by verifying the correct meaning may lead to better retention than simply inferring the target words' meanings because the combination of guessing and verifying may lead to deeper engagement with the lexical items than guessing alone. However, Mondria [63] found no advantage for guessing-plus-verifying compared to dictionary consultation in terms of learning, but guessing-plus-verifying took reliably longer than just dictionary consultation, suggesting that guessing-plus-verifying is less efficient than dictionary consultation alone. Furthermore, guessing and verifying prevents participants from guessing incorrectly and potentially learning incorrect meanings for lexical items. In fact, the current results are fully compatible with the idea that guessing can lead to no learning and possibly even incorrect learning. We found that participants who were better at guessing the correct meanings of target words during the training also showed higher learning than participants who were less successful at guessing correctly during the training. The above result means that, on the flip-side, participants who more frequently guessed incorrectly during training showed less learning, and the reason for this may be that the incorrect guesses during training lead them to learn an incorrect meaning for some items.

The present finding raises questions about what skills are needed for successful inferencing and whether strategies should explicitly be taught or should be adopted naturally in the language learner. Based on their results, Shangarfam et al. [17] argue that inferencing strategies should be explicitly instructed. However, inferencing requires the application of many processing strategies, including the use of extra-textual clues. That is, a text may not always provide enough information to infer the correct meaning and one has to rely on one's own knowledge for successful inferencing [96]. In the current study, participants were not explicitly taught guessing strategies, but were simply told to guess in order to let them naturally apply these techniques.

## Factors influencing word learning

Our third research question (RQ3) explored various factors that may influence the vocabulary learning effect when participants are guessing from context or looking words up in a dictionary. As our results suggest that vocabulary size influences word learning in various ways, we begin by discussing the implications of using a vocabulary size test involving self-report rather than a demonstration of knowledge, such that learners' personal characteristics may have influenced their vocabulary size score in the current study. For example, some learners may have ticked a word only if they are absolutely sure that they knew both the word form and meaning, whereas others may have ticked words that simply looked familiar to them. The results from the XK_Lex may thus reflect personality traits in addition to different vocabulary sizes. To what extent the scores derived by the XK_Lex are influenced by differences in personalities is not entirely clear. Studies in a US context have found that self-reported L1 vocabulary knowledge is either highly correlated with actual vocabulary knowledge [97] or that speakers moderately overestimate their vocabulary knowledge [98]. However, there are cross-cultural differences in accurately estimating and overestimating one's knowledge. For example, Vonkova et al. [99] found that students in East Asia estimate their knowledge highly accurately, but students in Southern and Central America do not. Furthermore, students in Southern Europe tend to exaggerate their knowledge more than those in Western Europe. However, there seems to be no consistent pattern in terms of accurately estimating and overestimating knowledge across the Middle East. Since Saudi Arabia is not included in Vonkova et al.'s study [99], it is not clear how students with this particular cultural background would fare in terms of accuracy and exaggeration of estimated knowledge.

Our own results from the training sessions, where we measured self-reported and actual knowledge of known target words, suggest that learners in the current study overestimated their knowledge of target words to a small to moderate extent. We additionally found that self-reported and actual knowledge of known target words was highly correlated. This shows that while participants over-reported their knowledge of target words during training, their reporting is highly consistent, namely consistently somewhat higher than their actual knowledge of target words. It seems reasonable to suggest that participants may have been equally consistent in over-reporting word knowledge in the XK_Lex. We would therefore like to suggest that participants may have overestimated their vocabulary knowledge in the XK_Lex test, which may have led to somewhat inflated vocabulary sizes in the current study, but that any possible effects of personality traits on the results of the XK_Lex are likely to be small. Nevertheless, this possible limitation of the XK_Lex test should be kept in mind in the following discussion.

First, we found that learners with a higher vocabulary size showed a larger vocabulary learning effect than learners with a lower vocabulary size in the guessing condition. In addition, participants with a larger vocabulary size were significantly more successful at guessing from context than participants with a lower vocabulary size. These results are in line with some previous studies [38,42] and with the claim that vocabulary knowledge plays an important role in language acquisition (e.g. [8,100–102]). This finding also supports the notion that substantial prior vocabulary knowledge is needed to guess meaning from context correctly and efficiently. Specifically, Huckin and Coady [103] suggested that in order to guess meaning successfully from context, a learner must have high lexical coverage so as to identify most of the surrounding lexical items. Similarly, Nation [8] claims that successful inferencing relies on learners' lexical knowledge. Higher vocabulary size eases the inferencing attempt as the more vocabulary learners know, the more text coverage they have and the more effective their guessing will be.

However, our results suggest that text coverage of less than 98% [31,33] may be sufficient for successful inferencing. Specifically, we propose that our participants' average vocabulary

size likely corresponded to only about 95% text coverage, which likely leads to adequate text comprehension only in a minority of cases [31,94]. While it is not entirely clear how participants' vocabulary knowledge, which is calculated in lemmas in the current study, translates into word families, we can estimate a reasonable range. Specifically, Lextutor's [86] vocabulary profiler determined that our texts contained between 1.10 and 1.18 word types per word family. This means that the 3000 word families needed for 95% text coverage for our texts correspond to a vocabulary size ranging anywhere from 3000 to 3540 (i.e. 3000 x 1.18) lemmas. Participants' actual average vocabulary size of 3331 lemmas is within this range. In contrast, the 4000 word families needed for 98% text coverage convert to a vocabulary size within the range of 4000 to 4720 (i.e. 4000 x 1.18) lemmas, which is above the average vocabulary size in the current study. While the above estimates give us a reasonable range of how word families relate to lemmas for the texts used in this study, we should also point out that the relationship between lemma knowledge and word family knowledge is not straightforward. Studies suggest an average number of 4 members per word family for English [80,104]. However, knowing one member of a family does not equate knowledge of all family members. For example, Kremmel and Schmitt's [105] data suggest that knowing the meaning of a base word allows connecting other word family members to the base word in only about 73% of cases [80]. Keeping these caveats in mind, we tentatively suggest that our participants likely had about 95% text coverage, but on average did not reach 98% text coverage. Thus, while we did find that participants with larger vocabulary size and thus likely higher text coverage for the particular texts used here learned more effectively through inferencing and were more successful at guessing correctly, participants in the current study overall likely had an average vocabulary knowledge below that suggested for successful inferencing. Our results thus confirm the importance of text coverage for successful inferencing, but suggest that in some cases learners may be able to successfully engage in lexical inferencing despite not reaching 98% text coverage.

It is possible that learners in the current study overall could engage in successful inferencing despite their relatively low vocabulary size because we chose reading texts from textbooks geared towards their level of knowledge. These texts may include cues that allow even learners with less text coverage to engage in successful inferencing. To tentatively explore this idea, we probed which target words were learned most successfully during the study and which cues might have contributed to this. Two patterns emerged: First, participants seemed to have particular trouble with words that appeared in a list, as in *Cognitive processes include perception, thinking, problem-solving, memory, language and attention* or *At least until the start of schooling, the family is responsible for teaching children cultural values, attitudes, and prejudices about themselves and others*. Lists do not provide any specific cues as to how the words listed are related, which may explain why target words found in lists were comparatively difficult to learn. In addition, unfamiliar words contained in lists may not be of high importance for text comprehension as a whole. Some of the words that were learned most successfully were attributive adjectives, such as *diverse issues* or *glamorous magazine*, and adverbs, as in *The iceman's hair was neatly cut* or *[S]ociology was born out of a concern with this rapidly changing character of the modern, industrial world*. Here, the meaning of both adjectives and adverbs is constrained by the nouns and verbs that they modify. For example, in the iceman example above, *neatly* modifies *cut* in the context of *hair*, which constrains the possible meanings for *neatly* to a manner of hair having been cut. Such constraints can aid guessing (as well as dictionary consultation) and act as cues to support the vocabulary acquisition process. In contrast, nouns modified by an adjective, as in *Cognition is based on a person's mental representations of the world, such as images, words, and concepts*, tended to yield moderate learning gains. It is

possible that adjectives do not constrain the meaning of the nouns that they modify as much as nouns constrain the meaning of the adjectives that can modify them.

Second, we found that learners with a higher vocabulary size showed a larger vocabulary learning effect than learners with a lower vocabulary size not only in the inferencing condition, but also in the dictionary condition. This novel result expands on previous findings as it highlights that successful vocabulary learning through dictionary use also relies on learners' vocabulary size. It seems that even when engaged in dictionary consultation, a strategy that even beginning learners can use successfully, a solid vocabulary base supports vocabulary learning. It is possible that learners with larger vocabularies could integrate and connect the words that they looked up more easily into their mental lexicon than learners with smaller vocabularies. This result thus puts into perspective Krashen's [106] idea that dictionary consultation is especially suited for novice L2 learners. Overall, our results support the idea that vocabulary knowledge is important for a wide range of tasks, even for such simple tasks as looking up words in a dictionary.

We further found that the more words participants already knew during training, the lower their learning effect in the guessing condition. This effect is most likely simply related to the experimental design. Participants who already knew many of the twelve target words that they were asked to guess during training could not learn many words. For example, a participant who already knew six of the twelve words, could only learn six words through guessing. In contrast, a participant who knew none of the twelve words, could potentially learn all twelve words through guessing. While our norming study ensured that participants overall knew few of the target words that they would be asked to learn during training, there were individual differences among learners that were beyond our control.

Finally, we found that the better participants were at guessing correctly, the better they learned the words in the guessing condition. Conversely, participants who were less successful as guessing the correct meaning, showed less learning. Our results therefore support the idea that guessing is risky because incorrect guesses may lead to learning incorrect meanings for vocabulary items. This result is in line with Huckin and Coady's [103] observation that inferencing is an imprecise technique, which may be problematic if a reading task asks for precise meanings. In the current study, participants who were less successful at guessing correctly may not have been entirely wrong, but may have guessed a lexical item related to the target translation. Interestingly, Mondria [63] found that words that were incorrectly inferred prior to verification were retained better than words that were correctly inferred before verification. This suggests that incorrect inferences are risky, but the process of discovering that a guess was incorrect seems to be beneficial for learning.

## Conclusion

In conclusion, the results from the current study suggest that both lexical inferencing and dictionary consultation led to substantial vocabulary learning over the course of the study. A larger vocabulary size at the beginning of the study supported acquisition of words trained through lexical inferencing, but also through dictionary consultation. The amount of learning of the particular words trained was thus influenced by participants' prior vocabulary size. In addition, learning through inferencing was also influenced by previous knowledge of the target words and how successful learners were at guessing correctly.

## Supporting information

**S1 Appendix.**
(DOCX)

## Acknowledgments

We would like to thank all the students who participated in the study. We would also like to thank the university instructors for allowing the first author to collect data during their class time and the Arabic specialist who assisted in the coding of the data. The first author is thankful for the technical support provided by Dr. Ibrahim AlKhazi.

## Author Contributions

**Conceptualization:** Alaa Alahmadi, Anouschka Foltz.

**Data curation:** Alaa Alahmadi, Anouschka Foltz.

**Formal analysis:** Alaa Alahmadi, Anouschka Foltz.

**Funding acquisition:** Alaa Alahmadi, Anouschka Foltz.

**Investigation:** Alaa Alahmadi.

**Methodology:** Alaa Alahmadi, Anouschka Foltz.

**Project administration:** Alaa Alahmadi.

**Supervision:** Anouschka Foltz.

**Visualization:** Alaa Alahmadi, Anouschka Foltz.

**Writing – original draft:** Alaa Alahmadi.

**Writing – review & editing:** Anouschka Foltz.

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
