## [Decision Letter · Decision Letter 0]

24 Feb 2020

PONE-D-19-34526

Exploring the effect of lexical inferencing and lexical translation on undergraduate EFL students’ vocabulary acquisition

PLOS ONE

Dear Dr. Foltz,

Thank you for submitting your manuscript to PLOS ONE. After careful consideration, we feel that it has merit but does not fully meet PLOS ONE’s publication criteria as it currently stands. Therefore, we invite you to submit a revised version of the manuscript that addresses the points raised during the review process.

I am sorry to have kept you waiting. Your manuscript (PONE-D-19-34526) was reviewed by two experts on language learning, and their comments are appended at the bottom of this letter. As you will see, they saw some merits in your study but also provided several major concerns. As an academic editor, I have read the manuscript myself. My view on the current version of your manuscript largely overlaps with the reviewers'.

Both Reviewers stated that the vocabulary size increase addressed under the research questions 4 and 5 is not easily acceptable. They both pointed out that, as you yourself mentioned in the manuscript, it might be merely due to the fact the learners took the same test twice. If you choose to revise your manuscript, I strongly request that you focus on the research questions 1, 2, and 3 (and remove 4 and 5). The two reviewers also request that some methodological details (e.g., target words, texts, dictionary) should be provided. I agree.

Given Reviewer 1's suggestion to request "Minor Revision" and Reviewer 2's suggestion to "Reject," my editorial decision here is "Major Revision." If and only if you find it possible to satisfy the reviewers' and my requests, please revise and resubmit your manuscript. Please note that this does not guarantee eventual acceptance of your manuscript. If resubmitted, depending on the quality of the revised manuscript, I might try to send it to the same reviewers or reject it right away at the editorial stage. Whether or not you resubmit to PLOS ONE, I hope that you will find the present reviews beneficial to your study.

Finally, there are some more minor issues:

line 52: VLS strategies

-> VLS (because S stands for strategy)

line 222: Likert scale always

-> scale: always

line 208: relevant for the current

-> relevant for the current ... what?

lines 243, 247: "family = poisson"

-> These analyses are not for count data. Please reconsider or justify your analyses.

line 280: "used as a general estimate of proficiency"

-> What kind of proficiency are you talking about here? Did you use GPAs as a general estimate of language proficiency?

line 303: "at the same time"

-> This sounds slightly misleading. What about "within the same session"?

line 348: "Participants' average ....

-> I found this line slightly awkward and difficult to understand.  You are talking about the difference in proficiency... are you?

We would appreciate receiving your revised manuscript by Apr 09 2020 11:59PM. To enhance the reproducibility of your results, we recommend that if applicable you deposit your laboratory protocols in protocols.io, where a protocol can be assigned its own identifier (DOI) such that it can be cited independently in the future. For instructions see: http://journals.plos.org/plosone/s/submission-guidelines#loc-laboratory-protocols

We look forward to receiving your revised manuscript.

Kind regards,

Koji Miwa, Ph.D.

Academic Editor

PLOS ONE

Journal Requirements:

2. Your ethics statement must appear in the Methods section of your manuscript. If your ethics statement is written in any section besides the Methods, please move it to the Methods section and delete it from any other section. Please also ensure that your ethics statement is included in your manuscript, as the ethics section of your online submission will not be published alongside your manuscript.

Reviewers' comments:

Reviewer's Responses to Questions

**Comments to the Author**

1. Is the manuscript technically sound, and do the data support the conclusions?

Reviewer #1: Yes

Reviewer #2: Partly

2. Has the statistical analysis been performed appropriately and rigorously? 

Reviewer #1: Yes

Reviewer #2: Yes

3. Have the authors made all data underlying the findings in their manuscript fully available?

Reviewer #1: Yes

Reviewer #2: Yes

4. Is the manuscript presented in an intelligible fashion and written in standard English?

Reviewer #1: Yes

Reviewer #2: Yes

5. Review Comments to the Author

Reviewer #1: Review of PONE-D-19-34526

Overall I find the study well designed and executed and the paper well-presented and thorough. I find the research questions to be answered appropriately by the method and analysis, though I am a little more skeptical than the authors about the reasons for increased vocabulary size over the duration. My hunch is that it is primarily a practice effect from taking the same test twice. I feel this makes RQs 4 and 5 somewhat unnecessary. Nevertheless, the authors’ approach to these questions and the discussion that they provide is interesting and well supported. Therefore, I recommend minor revisions but would suggest some more hedging and consideration of the counter-arguments regarding the increase in vocabulary size.

Major comments

My major concern reading the paper was the increase in vocabulary size (addressed in RQ4 and RQ5). The average increase of 506 words over five weeks is barely believable in terms of normal second language word learning, but when considering that this increase is actually derived from an increase in the average score of just five words (5.06 words, without considering distractors), it becomes more realistic. The authors provide a very well-researched and argued discussion (line 648~) of this finding, which makes the finding reasonably palatable. Nevertheless, it requires a little more explanation. At present, only one sentence mentions “Finally, participants completed the same test twice within a period of five weeks, which may have somewhat inflated the scores of the second test” (lines 668-669). To me, this is what I assumed while reading the paper, from the RQs onwards. Basically, I was puzzled why vocabulary size was measured twice in such a short duration using exactly the same test – it seems obvious that the test itself would play some role in learning, especially if it contains the same items. The first test provides an opportunity to notice words that are only partly known or completely unknown, followed by potential opportunities to learn these words during the learning period, resulting in an overall increase in vocabulary size on the second test. This narrative does seem to fit with the research though – that is, focused training on vocabulary learning should in fact provide this kind of learning opportunity. Nevertheless, I think one sentence does not quite do justice to the most likely reason why such a large increase in vocabulary size was observed, that is, a practice effect.

Also, on re-reading the Method section I noticed on line 276 that the formula may partly explain the increase in vocabulary size over the duration. Incorrect guessing (i.e., selecting pseudowords) is penalized in the formula; therefore, it is plausible that the increase in vocabulary size over the five weeks may also be partly due to reduced guessing (or increased awareness of) pseudowords the second time that participants took the test. Basically, this adds weight to the counter-argument that the increase in vocabulary size was due to a practice effect of taking the same test twice, rather than a real increase in word knowledge over the period.

Minor points

- line 89 - change ‘brain” to ‘lexicon’

- Could the target/control words be included as an appendix?

- Were any of the items borrowed words (loanwords) sharing form and meaning in Arabic and English? I assume not, but this should be mentioned at some point when describing the materials because cognates/loanwords are often easier to learn than non-cognates/non-loanwords.

- Replace ‘mean’ with M, ‘estimate’ with β, and ‘std.error’ with SE in the text?

- Line 231 - why was a GLM with quasipoisson distribution used instead of a non-parametric two-group test (e.g. wilcoxon test)?

- L243 - as above, why was a GLM with Poisson distribution used instead of t-test? I think a sentence should be provided explaining use of GLMs for these two-group comparisons.

- L266-268 this sentence needs correcting for grammar

- L307 - avoid contractions

- L339 - ‘test session’ should be ‘training session’ I think

- L363 - participants knew M=9.2 of 24 target words before training. Describing this as a ‘minority’ is true, but it is still a large proportion of target words that cannot be learned because they are already known.

Reviewer #2: This study follows a within-participants experimental design to analyse and compare the effectiveness of two vocabulary learning strategies, i.e., lexical inferencing and dictionary consultation, in the vocabulary development of undergraduate EFL learners. The participants had to learn a total of 24 target words using these two vocabulary strategies (12 words in each) over two training sessions. After the treatment, participants had to complete a translation task including the 24 target words and 24 control words, a vocabulary size checklist and a vocabulary strategies questionnaire. Generalised linear models showed that participants’ knowledge of the target words improved to a similar extent under both learning conditions, suggesting that the two strategies have a comparable learning effect on vocabulary development. This learning was also affected by the participants’ vocabulary size, with larger breadth of vocabulary leading to larger learning gains in both conditions. The authors also claim that learners’ vocabulary size increased as a result of the treatment and discuss the factors that might have led to this increase.

This study is interesting and generally well though through, and there is certainly merit in examining the effect of these two very popular learning strategies in L2 vocabulary development. However, the manuscript in its current form does not meet the quality requirements to warrant publication in PLOS ONE. The paper attempts to address many research questions, which consequently has affected the coherence of the paper and the discussion of the main aim. More importantly, there are certain methodological issues that required more rigorous control and consideration and have affected the quality of the results and the discussion.

These and other issues are described below in more detail, along with comments and suggestions on how they might be addressed in order to warrant publication in future attempts.

See attached pdf for more comments.

6. PLOS authors have the option to publish the peer review history of their article (what does this mean?). If published, this will include your full peer review and any attached files.

Reviewer #1: No

Reviewer #2: No

---

## [Author Response · Author response to Decision Letter 0]

9 Apr 2020

Please see the attached "Response to Reviewers" Word document for our detailed responses to the specific reviewer and editor comments. Please note that the document shows reviewer and editor comments in regular font and our responses in italics.

---

## [Decision Letter · Decision Letter 1]

18 May 2020

PONE-D-19-34526R1

Exploring the effect of lexical inferencing and dictionary consultation on undergraduate EFL students’ vocabulary acquisition

PLOS ONE

Dear Dr. Foltz,

Thank you for submitting your manuscript to PLOS ONE. After careful consideration, we feel that it has merit but does not fully meet PLOS ONE’s publication criteria as it currently stands. Therefore, we invite you to submit a revised version of the manuscript that addresses the points raised during the review process.

Your manuscript (PONE-D-19-34526_R1) was reviewed by the same two experts from the previous round. As you will see below, both reviewers commented that you revised the manuscript well to the extent that it is now almost ready for publication. I agree with both reviewers. I found the manuscript easier to read, with the story revised properly and necessary information supplied sufficiently. 

Although your journey is almost coming to the end, I would like to direct your attention to further suggestions provided by Reviewer 2. I find the reviewer's concern important. In your revision, I request that you respond to Reviewer 2's comments fully and revise the text appropriately. 

We would appreciate receiving your revised manuscript by Jul 02 2020 11:59PM. To enhance the reproducibility of your results, we recommend that if applicable you deposit your laboratory protocols in protocols.io, where a protocol can be assigned its own identifier (DOI) such that it can be cited independently in the future. For instructions see: http://journals.plos.org/plosone/s/submission-guidelines#loc-laboratory-protocols

We look forward to receiving your revised manuscript.

Kind regards,

Koji Miwa, Ph.D.

Academic Editor

PLOS ONE

Reviewers' comments:

Reviewer's Responses to Questions

**Comments to the Author**

1. If the authors have adequately addressed your comments raised in a previous round of review and you feel that this manuscript is now acceptable for publication, you may indicate that here to bypass the “Comments to the Author” section, enter your conflict of interest statement in the “Confidential to Editor” section, and submit your "Accept" recommendation.

Reviewer #1: All comments have been addressed

Reviewer #2: All comments have been addressed

2. Is the manuscript technically sound, and do the data support the conclusions?

Reviewer #1: Yes

Reviewer #2: Yes

3. Has the statistical analysis been performed appropriately and rigorously? 

Reviewer #1: Yes

Reviewer #2: Yes

4. Have the authors made all data underlying the findings in their manuscript fully available?

Reviewer #1: Yes

Reviewer #2: Yes

5. Is the manuscript presented in an intelligible fashion and written in standard English?

Reviewer #1: Yes

Reviewer #2: Yes

6. Review Comments to the Author

Reviewer #1: The authors have comprehensively addressed both reviewers' comments and have made significant revisions that have improved the manuscript considerably. I am happy to recommend that this paper be accepted. There are just a handful of minor textual clarifications that I believe need addressing to finalise the paper for publication.

1) I think a little reworking of the text is needed from lines 107 and 720. In these sections, the authors discuss the estimated lexical knowledge required for successful inferencing, citing Nation, who suggests 98% coverage needed for lexical inferencing, and Laufer & Ravenhorst-Kalovski, who say knowledge of 8-9000 words needed to achieve this coverage. Were these researchers were talking about a specific type of text, that is, academic texts? In other words, to read typical academic texts, knowledge of 8-9000 words is required to reach 98% coverage. The authors do not mention the genre but should do so in the section starting line 107.

Also, in the discussion from line 720 the authors suggest that 98% coverage is perhaps not required for successful inferencing because the participants were successful even though they had average vocabulary sizes of 3-4000 words, not 8-9000 words. But then from line 730 (and also from 429 where the texts are described in the methodology) the authors suggest that the texts used in this study were most likely simpler than typical academic texts, which is why participants could achieve successful inferencing. It is unclear to me whether the 98% coverage at 8-9000 words really applies in this case. Basically 98% coverage depends specifically on the lexis present in the texts and the learners’ lexical knowledge (a beginner learner can reach 98% coverage of some children’s books). Without reporting the specific number of words and their frequencies in the texts, it is difficult to convincingly challenge the claims by previous researchers.

The argumentation in these sections should therefore be clarified.

2) I found the following text (from line 690) somewhat misleading as well: “the current results are fully compatible with the idea that just guessing can lead to incorrect learning as we found that participant [-s missing here] who were better at guessing the correct meanings of target words during the training also showed higher learning than participants who were less successful at guessing correctly during the training.”

It seems to me that the evidence does not suggest that ‘guessing can lead to incorrect learning’ but instead suggests that ‘guessing can lead to no learning’. The authors would may wish to modify the text to resolve this issue.

3) From line 695: “strategies should explicitly be taught or should arise naturally in the language learner” – I think the latter part needs changing to “…or should be adopted naturally’ or similar. It seems odd to say strategies arise in someone.

Reviewer #2: See attached document.

I would like to commend the authors for their huge efforts to review and rewrite this manuscript. I feel that these have improved the paper considerably.

7. PLOS authors have the option to publish the peer review history of their article (what does this mean?). If published, this will include your full peer review and any attached files.

Reviewer #1: No

Reviewer #2: No

---

## [Author Response · Author response to Decision Letter 1]

29 Jun 2020

Please refer to the uploaded "response to reviewers" document.

---

## [Editor Report · Decision Letter 2]

15 Jul 2020

Exploring the effect of lexical inferencing and dictionary consultation on undergraduate EFL students’ vocabulary acquisition

PONE-D-19-34526R2

Dear Dr. Foltz,

We’re pleased to inform you that your manuscript has been judged scientifically suitable for publication and will be formally accepted for publication once it meets all outstanding technical requirements.

Kind regards,

Koji Miwa, Ph.D.

Academic Editor

PLOS ONE
---

## [Editor Report · Acceptance letter]

17 Jul 2020

PONE-D-19-34526R2 

Exploring the effect of lexical inferencing and dictionary consultation on undergraduate EFL students’ vocabulary acquisition 

Dear Dr. Foltz:

I'm pleased to inform you that your manuscript has been deemed suitable for publication in PLOS ONE. Congratulations! Your manuscript is now with our production department. 

Kind regards, 

on behalf of

Dr. Koji Miwa 

Academic Editor

PLOS ONE